# MicroRNA164 Affects Plant Responses to UV Radiation in Perennial Ryegrass

**DOI:** 10.3390/plants13091242

**Published:** 2024-04-30

**Authors:** Chang Xu, Xin Huang, Ning Ma, Yanrong Liu, Aijiao Xu, Xunzhong Zhang, Dayong Li, Yue Li, Wanjun Zhang, Kehua Wang

**Affiliations:** 1College of Grassland Science and Technology, China Agricultural University, Beijing 100193, China; bs20223241086@cau.edu.cn (C.X.); 16607128427@163.com (X.H.); 13581545284@163.com (N.M.); liuliyue2020@cau.edu.cn (Y.L.); s20203243162@cau.edu.cn (A.X.); liyue2020@cau.edu.cn (Y.L.); 2School of Plant and Environmental Sciences, Virginia Polytechnic Institute and State University, Blacksburg, VA 24061, USA; xuzhang@vt.edu; 3Beijing Academy of Agriculture and Forestry Sciences, Beijing 100097, China; lidayong@nercv.org

**Keywords:** UV-B radiation, total phenols, total anthocyanins, antioxidants

## Abstract

Increasing the ultraviolet radiation (UV) level, particularly UV-B due to damage to the stratospheric ozone layer by human activities, has huge negative effects on plant and animal metabolism. As a widely grown cool-season forage grass and turfgrass in the world, perennial ryegrass (*Lolium perenne*) is UV-B-sensitive. To study the effects of miR164, a highly conserved microRNA in plants, on perennial ryegrass under UV stress, both Os*miR164*a overexpression (OE164) and target mimicry (MIM164) transgenic perennial ryegrass plants were generated using agrobacterium-mediated transformation, and UV-B treatment (~600 μw cm^−2^) of 7 days was imposed. Morphological and physiological analysis showed that the *miR164* gene affected perennial ryegrass UV tolerance negatively, demonstrated by the more scorching leaves, higher leaf electrolyte leakage, and lower relative water content in OE164 than the WT and MIM164 plants after UV stress. The increased UV sensitivity could be partially due to the reduction in antioxidative capacity and the accumulation of anthocyanins. This study indicated the potential of targeting *miR164* and/or its targeted genes for the genetic manipulation of UV responses in forage grasses/turfgrasses; further research to reveal the molecular mechanism underlying how miR164 affects plant UV responses is needed.

## 1. Introduction

Ultraviolet radiation (UV) from the sun is divided into UV-A (400–320 nm), UV-B (320–280 nm), and UV-C (280–200 nm) according to the wavelength and biological effects. Although most higher plants and fungi would be damaged almost immediately from UV-C radiation, experiments show that if the amount of atmospheric ozone in the ozone layer reaches just 10% of the current level, it can still absorb all UV-C from the sun. UV-A radiation is basically harmless to plants, and even plays a regulatory role in the activities of plant life; however, despite the strong absorption of atmospheric ozone, there is still about 10% of UV-B radiation that reaches the ground. Its effect on plants is between UV-C and UV-A, and includes affecting plant growth and development and causing stress to plants [1]. Moreover, ozone depletion due to human activities resulted in an overall annual increase from 2 to 6% above the 1970–1980 UV levels after the 1990s, and this level will continue to increase in the future. This may pose a big threat to human health and global agricultural security as well [2,3]. 

As sessile organisms, plants have formed a series of morphological, physiological, biochemical, and molecular responses to adapt to different environmental changes during the long-term evolution process, including increased UV radiation. For instance, plants can adopt a number of protective strategies via the specific UV-B signaling pathways mediated by the UVR8 (UV RESISTANCE LOCUS 8) photoreceptor, such as increased leaf thickness, biosynthesis of UV-B reflective substances, increased antioxidants, and changes in UV-B-absorbing secondary metabolites at the cellular level [4,5,6]. By studying the different responses of plants under UV-B stress, we can further understand the mechanism of plant tolerance to UV-B and provide theoretical support for the future use of molecular biotechnology and genetic engineering methods to improve plant varieties.

MicroRNAs (miRNAs) are a class of conserved endogenous small non-coding RNAs with a length of 20–24 bases and they are key negative regulators in eukaryotic gene expression. Studies on miRNAs in *Arabidopsis thaliana*, maize (*Zea mays*), rice (*Oryza sativa*), and other model plants have shown that miRNAs play very important roles in the regulation of various aspects of plant growth and development, including leaf and flower morphogenesis, cell differentiation, tissue formation, hormone anabolism, signaling, and stress responses [7,8]. Their expression is generally tissue- and time-specific and is affected by various environmental stresses, such as salinity, drought, high temperature, and UV [9,10]. Global expression analysis of gene expression chips has helped researchers to speculate that 21 miRNAs from 11 miRNA families may be induced by UV-B radiation in *Arabidopsis* [11], and later experiments found 13 UV-upregulated and 11 UV-downregulated miRNAs [12] in *Populus tremula*, 17 highly UV-responsive miRNAs in maize leaves [9], and 7 UV-responsive miRNAs in wheat (*Triticum aestivum*) [13]; furthermore, high-throughput sequencing in grapes (*Vitis vinifera*) detected 13 differentially expressed miRNAs [14]. It has been found that the miRNAs that respond to UV stress in most plants are miR164, miR156, miR171, and miR398 [13,14,15,16], which indicates that these miRNAs may be involved in the regulation of targets after induction by UV-B stress.

The microRNA164 (miR164) family is a highly conserved class of miRNAs in plants [17]. Studies have found that the miR164 family in *Arabidopsis* consists of three members (miR164a, miR164b, and miR164c) which mediate five NAC (NAM, ATAF1/2, and CUC2) transcription factor genes; the miR164 family in rice has four mature sequences (miR164a/b/f, miR164c, miR164d, and miR164e), and the main targets are six NAC transcription factors. These target genes are mainly involved in the formation of vegetative and floral organs, the maintenance of organ boundaries, lateral root formation, senescence-mediated cell death, and biotic and abiotic stresses [18,19]. However, other than a few studies that showed that the expression of miR164 in plants changed under UV-B treatment [9,12,13], no report about the relationship between miR164 and the plant response to UV has been found so far, and the possible specific responses and tolerance mechanism need to be further revealed. 

Therefore, the objective of this study was to investigate how miR164 would affect responses to UV in perennial ryegrass (*Lolium perenne*), a UV-B-sensitive and widely grown cool-season turfgrass and forage grass species in the world [20]. The finding could provide some useful knowledge about the role of miR164 in plant response to UV radiation, perhaps leading to a better understanding of the mechanism of plant adaptation to changing UV environments. 

## 2. Results

### 2.1. Generation of Transgenic Perennial Ryegrass Plants with Overexpression or Target Mimicry of the Rice miR164a Gene

To investigate the function of the *miR164* gene in perennial ryegrass, an Os-*miR164a* overexpression or target mimicry construct (35S::Os-*miR164*/35S::Hyg, 35S::Os-*ISP1*-MIM164/35S::Hyg, Figure 1G) was constructed and introduced into perennial ryegrass by agrobacterium-mediated plant transformation (Figure 1A–F). A total of eight putative OE164 lines were identified using a genomic DNA template PCR for a 510 bp Os-*miR164a* fragment (Figure 1H) and five putative MIM164 lines were identified using PCR for a 257 bp Os-*ISP1*-MIM164 fragment (Figure 1I). Five independent OE164 or MIM164 lines were tested using semi-quantitative RT-PCR of the primary Os-*miR164* transcripts or Os-*ISP1*-MIM164 transcripts for further confirmation. All five of the selected OE164 lines were confirmed, but only one line of MIM164 was confirmed at the transcription level (Figure 1J,K). All five of the independent OE164 lines were morphologically similar and performed similarly when grown in the greenhouse. We selected one out of the five transcriptionally confirmed OE164 lines and one confirmed MIM164 line for further morphological and physiological analysis.

### 2.2. Phenotypic Characterization of Transgenic Plants

After comparing the OE164, WT, and MIM164 plants grown in the greenhouse, we observed that the OE164 plants were dwarfed, and had generally wider, shorter, and slower senescing leaves (fewer sensing leaves) than the WT plants, while it was nearly the opposite for MIM164 (Figure 2). 

### 2.3. MiR164 Affects Perennial Ryegrass Responses to UV Stress

#### 2.3.1. MiR164 Enhances Perennial Ryegrass Sensitivity to UV 

To understand the role of *miRNA164* in plant UV responses, the OE164, WT, and MIM164 plants were subjected to UV treatments for 7 days. The leaves showed a more severe wilting/scorching phenotype after 7 days of UV stress (UV+) in OE164 plants than those in both WT and MIN164 plants. And the MIM164 even looked slightly better than the WT plants (Figure 3A,B). To further investigate the different physiological changes among plants under UV+, we measured the leaf RWC and EL of OE164 and MIM164 in comparison with WT. UV caused a reduction in the RWC of all plants, regardless of the genotype. The result showed that OE164 had the lowest leaf RWC at both d3 and d7 of UV+, which indicated a worse water status in OE164 compared to both WT and MIM164 plants (Figure 3C). In accordance with the RWC, UV resulted in an increase in leaf EL, and the EL of OE164 was the highest among the plants after UV+, which was 80% and 25% higher than the WT and MIM164 plants at d3 and 43% and 22% higher than the WT and MIM164 plants at d7 of UV+, respectively (Figure 3D). These results indicate that UV-elicited leaf damage in OE164 plants was significantly more severe than that in the WT ones, while the damage in MIM164 was similar to that in the WT plants at d7, supporting a correlation with enhanced UV stress sensitivity in perennial ryegrass by miR164.

#### 2.3.2. Leaf Pigments and Total Phenolic Content

Here, the contents of leaf pigments (chlorophyll a, chlorophyll b, carotenoids, and anthocyanins) and total phenolic content were measured to further assess the effects of miR164 on perennial ryegrass responses to UV+. We found that the OE164 looked greener than both the WT and the MIM164, and further examination of the total chlorophyll contents showed that the OE164 had 31% and 37% higher chlorophyll contents than the WT and MIM164, respectively (Figure 4A,B). Generally, with an increase in the stress period, the plant chlorophyll a, chlorophyll b, and carotenoid contents showed a trend of decrease, while the total anthocyanin and phenolic contents presented an overall tendency to increase (Figure 4). Meanwhile, under normal growth conditions, the WT plant maintained higher total anthocyanin and phenolic contents and a higher chlorophyll a/b ratio than both the OE164 and MIM164 plants, whereas it had lower contents of chlorophyll a, chlorophyll b, and carotenoids than OE164. Three days after UV+, OE164 had higher chlorophyll b than both the WT and MIM164, higher carotenoids and total phenolic content than MIM164, and lower total anthocyanin and chlorophyll a/b ratio than the WT. There were no differences in all of the parameters among the plants at d7, except higher total anthocyanin and phenolic contents in the WT (Figure 4).

#### 2.3.3. Antioxidant Responses 

The activities of four major antioxidant enzymes, SOD, POD, CAT, and APX, were measured in perennial ryegrass after UV+. In general, UV+ increased the SOD activity and the OE164 plants had the highest activity among the three types of plants. For instance, the SOD activities of OE164, WT, and MIM164 at 3d of UV+ increased by 32%, 38%, and 57% when compared with those under the UV- condition, respectively (Figure 5A). POD activity also showed an overall increase in UV+, and OE164 had the lowest activity. Plant CAT activity showed an opposite response with a decrease after UV+ (Figure 5B). Similarly to the POD, OE164 had the lowest activity of CAT and APX among the three types of plants under UV+ (Figure 5C,D). Additionally, we further assayed the production of two major reactive oxygen species (ROS), O_2_^−^ and H_2_O_2_, in leaf tissues using NBT and DAB staining methods, respectively. The results showed that after UV+7d, OE164 had the strongest stain intensity for both O_2_^−^ and H_2_O_2_, while MIM164 showed the lightest stain intensity of O_2_^−^ and a similar stain intensity to the WT. Interestingly, under the control growth condition, OE164 had the lowest stain intensity of O_2_^−^ (Figure 5E,F).

## 3. Discussion

As a highly conserved class of miRNAs in plants, the microRNA164 (miR164) family has been widely studied in various plant species during the past (almost) 20 years [21,22,23,24,25,26,27,28]. Similarly to other miRNAs, miR164 functions in plant growth and development by forming regulation modules with its target genes, mainly NAC (NAM, ATAF, and CUC) transcription factors and a few less broadly conserved target genes containing a sequence complementary to miR164 with few mismatches. For instance, in *Arabidopsis* meristems, miR164 constrains the boundary domain expansion by degrading *CUC1* (*CUP-SHAPED COTYLEDON1*) and *CUC2* gene expression at the mRNA level [28]. Moreover, miR164-directed post-transcriptional regulation of *CUC1* is also required for the separation and normal development of adjacent embryonic, floral, and vegetative organs [29]. And in cotton (*Gossypium hirsutum*), miR164 forms a regulation module with GhCUC2-GhBRC1 to regulate plant architecture through abscisic acid [22]. In common wheat, miR164 regulates root growth and yield traits by targeting a phytosulfokine precursor gene, *TaPSK5* [30]. Here, phenotypic analysis found that OE164 had reduced plant height and delayed leaf senescence (greener and less senescing leaves) compared with the WT, while MIM164 was almost the opposite (Figure 2), which could be due to the downregulation of miR164-targeted *NAC* genes. Wang [31] reported that the knockout of miR164-targeted *OsCUC1* leads to multiple defects, including dwarf plant architecture in rice. Additionally, miR164 is involved in delaying leaf and fruit senescence by negatively regulating NAC transcription factor genes [32,33].

Besides the important roles in plant development, miR164 is also involved in regulating responses to biotic and abiotic stresses. Overexpressing a miR164-targeted NAC transcription factor NAC21/22 negatively regulates stripe rust resistance in wheat [34] and the overexpression of miR164-targeted NAC transcription factor genes has negative effects on drought resistance in rice at the reproduction stage [18]. Similarly, in wheat seedlings, tae-miR164 inhibits root development and reduces drought and salinity tolerance by downregulating the expression of *TaNAC14* [21]. In this study, we found that miR164 affected plant responses to UV by generating *OS-miR164a*-overexpressing (OE164) and -silencing (MIM164) perennial ryegrass, more specifically, by having a negative effect on UV tolerance. 

UV affects plant growth and development, and the type of plant responses to UV-B basically depend on the flux of UV radiation. At a low UV flux, it can promote metabolic and developmental changes, such as the biosynthesis of phenolic secondary metabolites and inhibition of hypocotyl elongation. UV-B radiation at a high flux can destroy leaf epidermal cells, even mesophyll cells, resulting in necrosis on the surface of the injured plant leaves [35]. UV+7d scorching and wilting leaves were observed in all three types of plants, particularly the OE164 plants. Furthermore, UV+ resulted in a reduction in the leaf RWC and an increase in the EL of all plants, regardless of the genotype (Figure 2). In accord with the leaf observation, OE164 had the highest EL and lowest RWC among the plants after UV+. UV-B radiation can also cause damage to the thylakoid membrane of plant cells, resulting in the production of ROS, mainly including superoxide anion radical (O_2_^−^), hydroxyl radical (·OH), hydrogen peroxide (H_2_O_2_), and singlet oxygen (^1^O_2_). ROS is very active in reacting with other macromolecules, such as proteins, lipids, nucleic acids, etc., resulting in damage to these substances and even cell death [1]. NBT and DAB staining of leaf tissues showed a higher production of both O_2_^−^ and H_2_O_2_ in OE164 than both WT and MIM164 plants after UV+7d, indicated by more severe oxidative stress in OE164, which was a good coincidence with the previous observations (i.e., leaf scorching, RWL, EL). 

In response to UV-B stress, plants have evolved specific UV-B signaling pathways to regulate growth, development, and protective adoption strategies [4,5]. During the growth and development of plants, the production and quenching of ROS in plants have dynamic balance. When plants are subjected to UV-B stress, a large number of ROS will be produced in plants. In order to reduce the damage caused by ROS, plants produce a series of anti-oxidation protection systems to remove ROS, which can often be divided into enzymatic protection systems and non-enzymatic protection systems. The enzymatic system mainly includes SOD, CAT, POD, APX, etc., while the non-enzymatic system not only includes ascorbic acid (ASA) and glutathione (GSH) in the Asada–Halliwell–Foyer cycle, but a number of molecules with radical scavenging ability can also provide the potential for ROS scavenging [36,37]. For example, plants contain a large number of non-enzymatic antioxidants, including polyphenols, carotenoids, polyamines, and proteins carrying redox-active thiol groups, etc., thus forming a dynamic network of redox interactions [38,39].

O_2_^−^ is dismutated to H_2_O_2_ by SOD, which is the first line of the enzymatic defense system against ROS. And H_2_O_2_ is then further detoxified by CAT and an array of peroxidases, such as APX and POD [40,41]. Under UV+, the activity of SOD increased, which indicated an adaptative response to the increase in oxidative damage on the plants by UV stress (Figure 5A). Interestingly, OE164 maintained higher SOD activity than both WT and MIM164 plants under both control and UV conditions; it had the lowest O_2_^−^ accumulation indicated by the NBT staining among the three types of plants concurrently under the control condition, but the highest O_2_^−^ accumulation 7 d after UV+ (Figure 5E,F). In general, the activities of these antioxidant enzymes increase concomitantly with a decrease in ROS production, or vice versa [40,42], but a stable antioxidant system status could depend on stress duration, intensity, and probably other factors [43]. For example, lily (*Lilium longiflorum*) showed that lower SOD activity was concurrent with greater O_2_^−^ production 10 h after heat stress, whereas higher SOD activity was accompanied by an equal or much higher O_2_^−^ concentration 6 or 8 h after heat stress was also observed [44]. We inferred that the higher O_2_^−^ accumulation in OE164 after UV stress was more likely an indication of much harsher oxidative stress damage than the WT and MIM164 plants, which would stimulate a higher SOD level in OE164 as a response in turn, or that there was just not high enough SOD to cope with the severity of the stress. The higher CAT, POD, and APX activities in MIM164 and WT over OE164 showed a higher capacity to detoxify the H_2_O_2_ under UV+, which paralleled the lower accumulation of H_2_O_2_ as demonstrated by the lighter stain intensity (Figure 5B–F). The reduced enzymatic antioxidant system, especially the lower H_2_O_2_ scavenging enzymes, could explain, at least partly, the enhanced UV sensitivity of OE plants.

In addition, the induction and synthesis of phenolic compounds (e.g., phenylpropanoids, flavonoids) is one of the most important and universal protective reactions of plants to UV radiation, as they act not only as a primary antioxidant to scavenge ROS, but also as a natural ‘UV sunscreen’ [45]. The accumulation of flavonoids (astilbin, quercetin, kaempferol, anthocyanins, etc.) and related UV-absorbing compounds in epidermal tissues reduces epidermal UV transmittance. And it is one of the main mechanisms for plants to adapt to UV environmental changes, including those caused by stratospheric ozone depletion and climate change [39,46]. The total phenolic content increased from 7.80 to 14.39, 12.09 to 15.63, and 8.34 to 11.91 mg/g DW in OE164, WT, and MIM164 after UV+7d, respectively. The OE164 and MIM164 had lower total phenols than the WT under UV-, but OE164 was not different from the WT both 3 and 7 d after UV stress. Similarly, UV+ also increased the total anthocyanin content, and both OE164 and MIM164 had lower total anthocyanin content than the WT plants (Figure 4E,F). This all indicated that the different accumulation of the total phenols, particularly the anthocyanins, might account for, to a certain degree, the difference in UV sensitivity among the three types of ryegrass plants. Different from the pigment anthocyanins, chlorophyll concentration usually decreases in response to high-level stresses, including UV [47,48,49]. All of the ryegrass plants showed a reduction in both the chlorophyll a and b concentrations after UV+, and OE164 maintained a lower chlorophyll a/chlorophyll b ratio (Chl a/b) than the WT and MIM164 for both UV- and UV+3d (Figure 4). The range of the Chl a/b ratio for healthy plants is between 2.5 and 3.5 and a higher Chl a/b ratio was reported to be a favorable trait in *Euglena gracilis* under shade [50] and in rice under oxidative stress [51]. However, whether lower Chl a/b in OE164 is related to plant UV sensitivity would need further detailed studies.

## 4. Conclusions

In summary, we generated both Os*miR164*a overexpression and target mimicry perennial ryegrass. Morphological and physiological analysis showed that the *miR164* gene affected perennial ryegrass UV tolerance in a negative way, demonstrated by the more scorching leaves, higher leaf LE, and lower RWL in OE164 than the WT and MIM164 plants. The increased UV sensitivity could be partially due to the reduction in antioxidative capacity and the accumulation of anthocyanins. This study indicated the potential of targeting miR164 and/or its targeted genes for genetic regulation of UV responses in forage grasses/turfgrasses, and further study to dissect the molecular mechanism through which miR164 affects plant UV responses is warranted.

## 5. Materials and Methods

### 5.1. Plant Materials, Growth Conditions, and Experiment Treatments

Perennial ryegrass “Citation Fore” (Pure Seed, Canby, OR, USA) was used in this study. MiR164-overexpressing (OE164) and target mimicry (MIM164) transgenic perennial ryegrass plants harboring the genes of interest under the of 35S promoter from the plant pathogen Cauliflower Mosaic Virus (CaMV) were developed as described previously [23,52,53]. Briefly, the OE164 contains pre-*OsmiR164a* (same mature miR164 sequence as perennial ryegrass, 5′-UGGAGAAGCAGGGCACGUGCA-3′) amplified by PCR using Nipponbare rice genomic DNA as the template. To generate a target mimicry construct (reducing the active level of miR164), *Arabidopsis AtISP1* (induced by phosphate starvation1), showing incomplete complementarity to *At-miR399*, was modified to mimic the targets *OsMIM164a* using DNA synthesis (General Biol, China); the resulting gene showed incomplete complementarity to the mature *OS-miR164a*, with four additional unpaired bases (TCTA) in the center [52,53,54] (Figure 1). After confirmation by Sanger sequencing, the correct sequences of *OS-miR164a* and the target mimicry sequence (*OsMIM164a*) were cloned into pZh01 and the pTCK303 vector with the hyg gene for hygromycin resistance as a selectable marker, respectively. The plasmid was then transferred into the *Agrobacterium tumefaciens* strain LBA4404 for plant transformation. The transformed *Agrobacterium* containing the overexpression construct of *OsmiR164a* and *OsMIM164a* was used to infect the embryonic callus induced from the mature seeds of perennial ryegrass, respectively. Transgenic plants were produced and maintained as described previously [55]. Target genes in putative overexpressing (OE164) and target mimicry (MIM164) lines in the T0 generation were identified by PCR and RT-PCR using sequence-specific primers (Table 1).

Wild-type (WT) control plants and one line of each transgenic perennial ryegrass plant (OE164, MIM164) were clonally propagated from tillers in plastic pots (8.5 cm depth and 10 cm diameter) filled with a soil mixture of sand/peat (1:1) in the greenhouse (15 ± 2 °C/25 ± 3 °C, night/day) at China Agricultural University (Beijing, China). The plants of the same growth status were selected and transferred to plant growth chambers (PQX-450, Shanghai Haixiang Instrument Factory, Shanghai, China) that provided LED white light, photosynthetically active radiation (PAR) of 400 µmol s^−1^ m^−2^, a dark/light cycle of 16/22 °C for 10/14 h, and 70% relative humidity. The plants were watered every other day, clipped to 10 cm, and fertilized weekly using Miracle-Gro (5 kg N ha^−1^ per month, N-P-K 24-12-14, Scotts) to achieve uniform plant growth. After acclimatizing to chamber conditions for 2 weeks, 30 pots of plants (5 pots/replication per treatment, UV− and UV+) were randomly subjected to UV radiation treatment (SANKYO DENKI G15T8E UV-B fluorescent tube, wavelength of 280–360 nm, peak at 302–306 nm; UV-AB 1000 μw cm^−2^, UV-A 400 μw cm^−2^) for 7 days (22/16 °C, 14/10 h, d/n, PAR 400 µmol s^−1^ m^−2^). In order to minimize the effects of the microenvironment, the pots in the chamber were rotated every 24 h during the experiments.

### 5.2. Sampling and Measurements

#### 5.2.1. Phenotypic Analysis of Transgenic Plants

Phenotypic assessment was conducted on WT and transgenic plants (OE164 and MIM164) after being transferred and maintained under normal growth conditions in a greenhouse for 4 months. The width of the ryegrass leaves was determined by measuring the width of the widest part of the first fully developed leaf (from the top to the bottom of the plant) of each tiller with a vernier caliper, and the same leaf was used for measuring the length from the leaf collar to the leaf tip. Ten measurements of randomly selected leaves were taken for each pot of plants. Plant height was measured from the soil surface to the top of the plants.

#### 5.2.2. Measurement of Physiological Parameters

Leaf chlorophylls and carotenoids were extracted using dimethyl sulfoxide (DMSO) and quantified as described previously [56]. The absorbance of extracts was read at 665, 649, and 480 nm using a UV-VIS spectrophotometer (Hitachi UH5300, Tokyo, Japan). Chlorophyll a, b, and carotenoids were calculated as milligrams per gram dry weight (DW) using the following equations: Chlorophyll a (Chl_a_) = 12.19 × A_665_ − 3.45 × A_649_, Chlorophyll b (Chl_b_) = 21.99 × A_649_ − 3.52 × A_665_, and Carotenoids = (1000 × A_480_ − 2.14 × Chl_a_-70.16 × Chl_b_)/220. Total anthocyanin content was quantified following the procedure described by Li et al. [57], with minor modifications. Briefly, 0.2 g of finely ground leaf powder in liquid nitrogen was mixed well with 10 mL acidic ethanol and incubated for 24 h at 4 °C in darkness with gentle shaking. The aqueous phase was then collected after centrifuging at 12,000× *g* for 8 min. The absorbance was subsequently measured at 530 nm using a UV-VIS spectrometer. Total anthocyanin content (TAC) was calculated using the following equation: Anthocyanin content (Anth) = A_530_ × V × N × 10^–2^ × 98.2^−1^ × M^–1^, where A_530_ is the absorbance at 530 nm, V is the volume of the extract (mL), N is the dilution factor, 98.2 is the molar absorptivity constant for the acidic ethanol at 530 nm, and M is the weight of the plant material used for extraction (g). Total phenolic content (TPC) determination of the Folin–Ciocalteu assay was conducted according to a method described previously [58,59], with minor modifications. Briefly, leaf tissues were ground into powder in liquid nitrogen. Then, 0.2 g powder was mixed with 10 mL of 50% acetone and sonicated in an ultrasonic bath for up to 10 min. The aqueous phase was collected after centrifuging at 10,000× *g* for 10 min. A total of 1 mL of diluted sample was added with 5 mL H_2_O, 1 mL Folin–C reagent, and 3 mL Na_2_CO_3_ solution (7.5% *w*/*v*), and then quickly mixed by shaking. We let the reaction mixture stand for 2 h at room temperature and the absorbance was subsequently measured at 765 nm. Gallic acid was used as the standard and total polyphenol contents were expressed as mg gallic acid per g dry weight.

Leaf relative water content (RWC) was measured in accordance with the method by Barrs and Weatherley [60], with minor modifications [61]. Leaves were harvested 3 and 7 d after UV treatments. About 10 fully expanded leaves (~0.2 g) from each plant pot were weighed (W1) and cut into 1–2 cm pieces. The leaf weight was measured (W2) after soaking in deionized distilled water (dd H_2_O) for 24 h. The leaves were then dried to constant weight (W3) in an oven at 65 °C. RWC (%) = (W1 − W3)/(W2 − W3) × 100%. Leaf electrolyte leakage (EL) was measured as described by Blum and Ebercon [62], with some modifications. Another 0.2 g of freshly harvested leaves was soaked into 20 mL dd H_2_O. The initial conductivity (S1) of leaf electrolyte leakage after shaking on a shaker for 24 h and the final conductivity (S2) after killing the leaves autoclaved for 20 min was measured using a conductivity meter (Hengxin TDS86555, Taiwan). EL = S1/S2 × 100%.

Enzyme extracts were prepared using the method by Chaitanya et al. [63], with modifications [43]. Leaf samples of 0.2 g were frozen quickly in liquid N and then ground in 4 mL solution containing 50 mM pH 7.8 phosphate buffer (PBS), 0.1 mM ethylenediaminetetraacetic acid (EDTA), and 1% polyvinylpolypyrrolidone (PVPP). The homogenate was centrifuged at 12,000× *g* for 15 min at 4 °C, and the supernatant was used for antioxidant enzyme activity assays. The activities of superoxide dismutase (SOD), ascorbate peroxidase (APX), guaiacol peroxidase (POD), and catalase (CAT) were determined following a previous protocol [61,63]. For measuring SOD activity, the reaction mixture without enzyme extract developed the maximum color after illumination and was used for maximum reduction of p-Nitro-Blue tetrazolium chloride (NBT); the mixture without illumination served as the control. The A_560_ was measured using a spectrophotometer. For measuring the activity of APX, the absorbance of a 3 mL reaction mixture containing 50 mM pH 7.0 PBS, 0.1 mM H_2_O_2_, 0.5 mM ASA (ascorbic acid), and 50 µL enzyme extract was recorded at 290 nm for 3 min. The A_470_ of a 3 mL assay solution (20 mM pH 6.0 PBS, 0.4 mM 2-methoxyphenol, 0.3 mM H_2_O_2_, and 0.2 mL enzyme extract) was recorded for 3 min to measure POD activity. The CAT activity was quantified by mixing 3 mL reaction solution containing 50 mM pH 7.0 PBS, 10 mM H_2_O_2_, and 30 μL enzyme extract. The A_240_ was recorded for 3 min. A unit of APX, POD, and CAT activity was defined as the quantity of enzyme that changes the absorbance (A_290_, A_470_, and A_240_, respectively) by 0.01/min under the specified conditions. Leaf H_2_O_2_ and superoxide production were detected by the DAB (3,3′-diaminobenzidine) staining method [64] and the NBT staining method [65], respectively. The staining intensity was further quantified using Quantity One 4.6.2 software (Bio-Rad, Hercules, CA, USA). 

### 5.3. Experimental Design and Statistical Analysis

The experiment consisted of a split-plot design with five replications. The UV treatments (control, UV-; UV-B radiation, UV+) were defined as the whole/main-plot and three genotypes of ryegrass plants (WT, OE164, MIM164) were defined as the split/sub-plot. We analyzed all of the measurements using the samples collected at the sampling time described previously (3 d and 7 d after UV treatments), unless stated otherwise. All of the data obtained were analyzed using SPSS software (Version 25.0, IBM, Armonk, NY, USA). Treatment means were separated using Fisher’s least significant difference (LSD) test at a significance level of 0.05.

## Figures and Tables

**Figure 1 plants-13-01242-f001:**
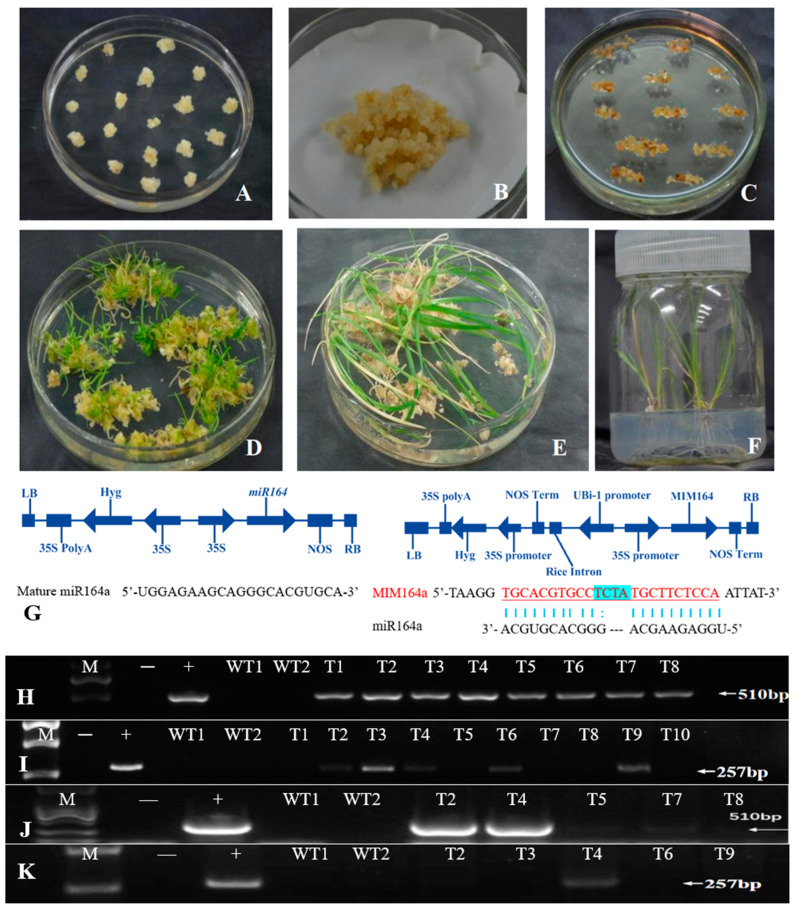
Generation and molecular confirmation of Os-miR164a overexpression (OE164) and target mimicry (MIM164) transgenic perennial ryegrass plants. (**A**–**F**) Callus transformation and transgenic plant generation process. (**G**) The schematic map of the T-DNA region of Os-miR164a overexpression (35S::Os-miR164//35S::Hyg) and target mimicry (35S::At-ISP1-MIM164//35S::Hyg) construct; UBi, maize ubiquitin promotor; 35S, CaMV 35S promoter; LB/RB, left/right border; NOS, nopaline synthase; PolyA, poly adenine; Hyg, hygromycin B phosphotransferase gene. (**H**,**I**) PCR assay of miRNA164 overexpression and target mimicry transgenic plants; T1–8/T1–10, putative OE164/MIM164 plants; +, positive control, pZH01. (**J**,**K**) RT-PCR assay of miRNA164 overexpression (**J**) and target mimicry (**K**) transgenic plants; T1–5/T1–5, putative OE164/MIM164 plants; +, positive control, pTCK303; −, negative control, H2O; WT, wild type; M, molecular marker.

**Figure 2 plants-13-01242-f002:**
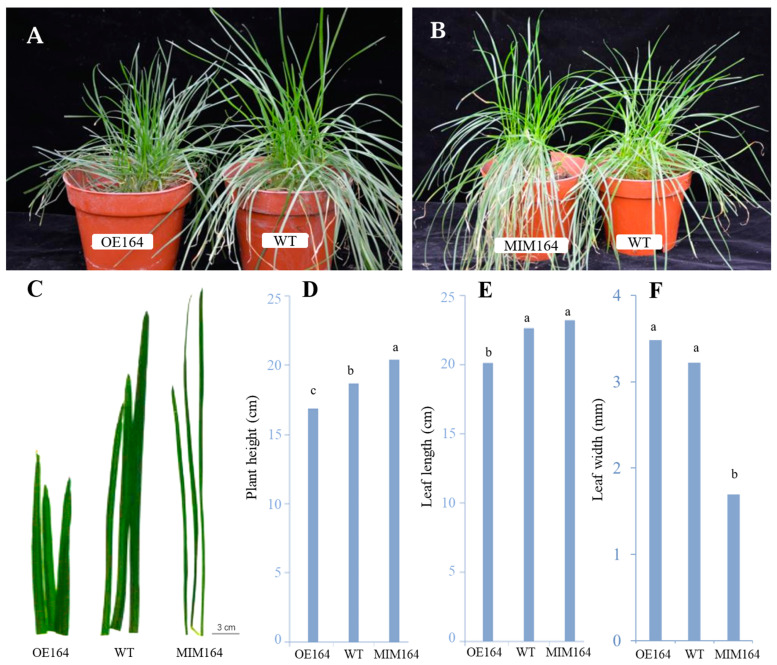
Phenotypic analysis of Os-*miR164a* overexpression (OE164) and target mimicry (MIM164) transgenic perennial ryegrass plants. (**A**) Comparison of whole plants of OE164 and WT. (**B**) Comparison of whole plants of MIM164 and WT. (**C**) Comparison of leaves of OE164, WT, and MIM164. (**D**–**F**) Plant height, leaf length, and leaf width of OE164, WT, and MIM164. The different lowercase letters indicate a significant difference among plants at *p* < 0.05, *n* = 40.

**Figure 3 plants-13-01242-f003:**
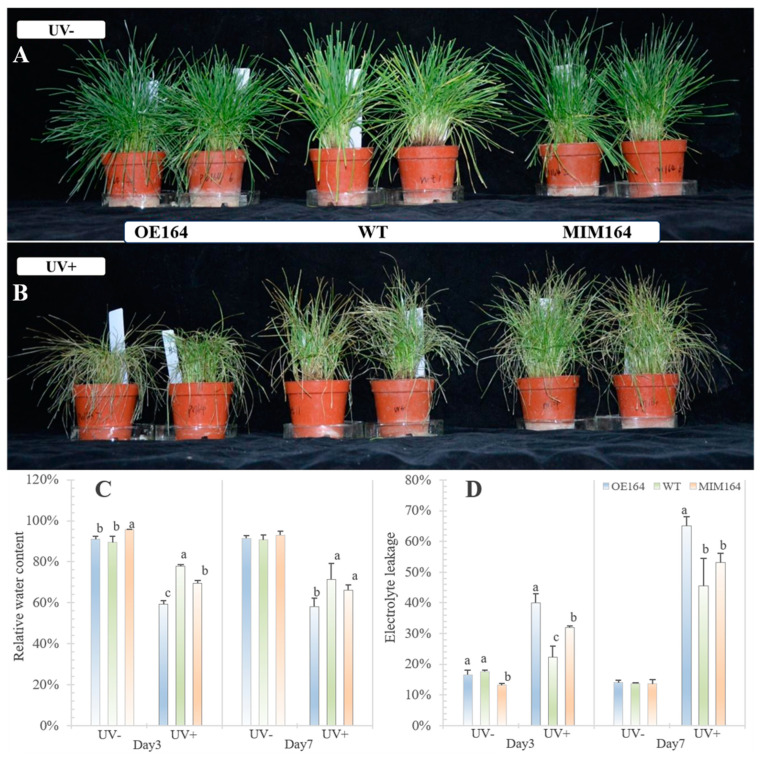
Morphological changes (**A**,**B**) and leaf relative water content (**C**); leaf electrolyte leakage (**D**) of OE164, WT, and MIM164 plants after UV+. The different lowercase letters indicate a significant difference among the plants 3 and 7 days after UV+ at *p* < 0.05, *n* = 5.

**Figure 4 plants-13-01242-f004:**
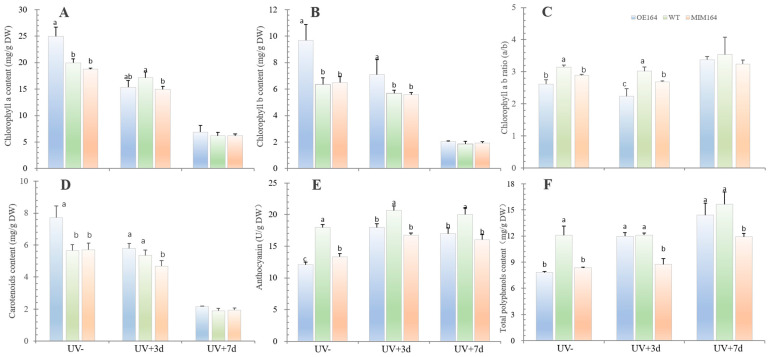
Leaf pigments and total phenolic content of OE164, WT, and MIM164 plants after UV+. (**A**–**C**) Chlorophyll a and b content and chlorophyll a/b ratio; (**D**) carotenoid content; (**E**) total anthocyanin content; (**F**) total phenolic content. UV−, normal growth condition. UV+3d and UV+7d 3 and 7 days after UV+. The different lowercase letters indicate a significant difference among the plants at *p* < 0.05, *n* = 5.

**Figure 5 plants-13-01242-f005:**
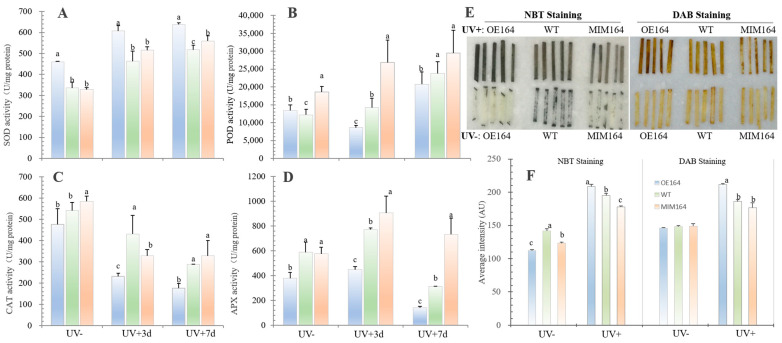
The activities of antioxidant enzymes and production of reactive oxygen species of OE164, WT, and MIM164 plants after UV stress. (**A**–**D**) The activities of superoxide dismutase (SOD), guaiacol peroxidase (POD), catalase (CAT), and ascorbate peroxidase (APX); (**E**,**F**) leaf NBT and DAB staining and staining intensity quantification. UV−, normal growth condition. UV+3d and UV+7d 3 and 7 days after UV treatment. The different lowercase letters indicate a significant difference among the plants at *p* < 0.05, *n* = 5.

**Table 1 plants-13-01242-t001:** Information on the primers used in this study.

Primer Name	Primer Sequence (5′–3′)	Purpose
**miR164-F-Xba**	TCTAGATTGCTTCAGTTGTTCGCAGT	Vector construction and PCR and RT-PCR analysis
**miR164-R-Sal**	GTCGACGCGTCTTAGTGTTACTTTGGAC	Vector construction and PCR and RT-PCR analysis
**ISP1-MIM164-F-Kpn**	TGGTACCAAAACACCACAAAAACA	MIM164 vector construction
**ISP1-MIM164-R-Spel**	ACTAGTAAGAGGAATTCACTATAAA	MIM164 vector construction
**MIM164-F**	GCCTAAATACAAAATGAAAACTCTC	PCR and RT-PCR analysis
**MIM164-R**	GTACAACCCAAACATAATGAAGAA	PCR and RT-PCR analysis

## Data Availability

The original contributions presented in the study are included in the article.

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
