# Peer review of "MicroRNA164 Affects Plant Responses to UV Radiation in Perennial Ryegrass"

_plants, 2024, doi:10.3390/plants13091242_

Round 1
Reviewer 1 Report
Comments and Suggestions for Authors
The study reported by Xu et al. shows that the rice miRNA164 negatively regulates UV stress tolerance in perennial ryegrass, possibly due to the reduction in anti-oxidative capacity and the accumulation of anthocyanins. Although the authors made great effort to generate transgenic lines, more evidence are needed to further support their conclusions. My suggestions are as follows:
1. The authors could test a few known miRNA164 target genes in the OE and RNAi lines to see whether the transgenic miRNA164 and knock-down are really effective, and to prove that the phenotypic results is indeed due to miRNA164.
2. Do the authors have more than one line of OE and MIM to support their results?
3. Is miRNA164 induced by UV in rice or perennial ryegrass? Expression profile of miRNA164 could be tested in rice or ryegrass.
4. The authors need to discuss why they can analyze the UV stress tolerant phenotype even though the OE and MIM already show difference under normal growth condition.
Author Response
请参阅附件

Reviewer 2 Report
Comments and Suggestions for Authors
In the manuscript by Xu et. al. entitled, “MicroRNA164 affects plant responses to UV radiation in perennial ryegrass”, the effect of miR164 was investigated under UV stress. This was performed through the generation of Os-miR164a overexpression (OE164) and knockdown (MIM164) constructs, followed by their introduction into perennial ryegrass via Agrobacterium-mediated plant transformation. Positive transformations in overexpression and RNAi lines were identified through PCR. Phenotypic analysis of the overexpression and knockdown lines revealed distinct morphological and physiological differences compared to wild-type (WT) plants. Overexpression lines exhibited dwarfism, wider and shorter leaves, and slower leaf senescence, whereas knockdown lines displayed opposite characteristics. Additionally, OE164 plants exhibited higher chlorophyll content and greener appearance compared to both WT and MIM164 plants. Under UV stress, OE164 plants showed more severe leaf wilting and scorching compared to WT and MIM164 plants, indicating increased sensitivity to UV stress. Furthermore, OE164 plants exhibited reduced leaf relative water content and increased leaf electrolyte leakage under UV stress, indicating compromised water status and cellular membrane integrity. Analysis of leaf pigments and total phenolic content revealed differential responses to UV stress among the plant lines, with OE164 plants showing unique pigment composition changes compared to WT and MIM164 plants. Additionally, the activities of antioxidant enzymes (SOD, POD, CAT, and APX) were assessed. OE164 plants showed increased production of reactive oxygen species compared to WT and MIM164 plants under UV stress, suggesting impaired ROS scavenging capacity in OE164 plants. Overall, these findings provide insights into the role of miR164 in perennial ryegrass responses to UV stress, with OE164 plants exhibiting increased sensitivity and compromised physiological responses compared to WT and MIM164 plants. This study highlights the importance of miR164 in regulating various aspects of plant growth, development, and stress responses in perennial ryegrass.
I have a few concerns that need to be addressed, mentioned as follows:
1. Quantification of miR164 targets (focusing on conserved genes of different species) including those related to oxidative stress should be performed to gain insights into the activity of miR164.
2. The figures can be made to look more professional and not out of proportion.
3. The superscription and subscription of letters or texts should be properly done.
4. The control/treatment can be mentioned as -UV / +UV for better clarity.
Comments on the Quality of English LanguageNone.
Author Response
请参阅附件

Reviewer 3 Report
Comments and Suggestions for Authors
This paper reports on the role of microRNA164 in the response of perennial ryegrass (Lolium perenne) plants to UV radiation. The authors show that miR164 enhances UV sensitivity and quantitatively analyze the effect of miR164 on the levels of chlorophylls a and b, carotenoids, anthocyanins, and total phenolic content, as well as the activity of antioxidant enzymes under UV irradiation conditions. The reported data are novel and interesting and undoubtedly worthy of publication in Plants. However, I have concerns that should be addressed before the paper can be accepted for publication.
Major points
1. The experiments were performed with uncharacterized transgenic lines. The authors show that the transgenic plants used indeed contained and expressed the introduced transgenes. However, it is not shown that these transgenes had any effect on the level/activity of miR164 in the plants. The authors assume that the activity of miR164 is increased and decreased in two transgenic lines used, but there is no experimental basis for this assumption. Additional experiments are needed to justify the assumed effect on miR164 activity. For "miR164-overexpressing" plants, miR164 levels can be directly compared to non-transformed plants, but this approach is not possible for MIM164 plants, as in this case activity rather than miR164 production is expected to be downregulated. Therefore, to demonstrate that the introduced transgenes have affected the activity of miR164, I would suggest using qPCR to measure the mRNA level for one or two known miR164 target mRNAs.
2. More than one transgenic line should be used to demonstrate the effect of a transgene.
Minor point
When describing MIM164 transgenic plants, the authors use the terms "RNAi plants" and "knockdown plants". Both are incorrect and misleading. In fact, these plants contain a transgene that directs the expression of an mRNA containing an uncleavable target of miR164, and this mRNA is expected to bind miR164, thus reducing the level of active miR164 molecules in the cell. Therefore, the authors should correct the wording used to describe MIM164 transgenic plants.
Round 2
Reviewer 1 Report
Comments and Suggestions for Authors
The authors claim that they have 5 independent OE lines. The data of these OE lines with similar results should be presented in the main figures.
Reviewer 2 Report
Comments and Suggestions for Authors
I appreciate the revisions made by the authors and recommend publishing the manuscript.
Author Response
Thanks!
Reviewer 3 Report
Comments and Suggestions for Authors
First of all, I would like to point out that it looks very strange when authors evaluate the reviewer's comments by calling them "wonderful," "great," "good," and so on. Instead of giving these ratings, authors should focus on the concerns raised by the reviewer.
Unfortunately, the authors have done nothing to provide adequate answers to two major concerns raised in the first review. In this situation, considering the impossibility of obtaining additional transgenic lines in a reasonable time, I would continue to insist only on experimental evidence that the introduced transgenes have the expected effects on miR164 activity. Therefore, it is necessary to find a way to quantify the mRNA level for one or two miR164 mRNA targets in two transgenic lines used relative to control non-transformed plants. If the time given for revising the paper is not sufficient to perform the required experiments, additional time can be requested from the editor.
Round 3
Reviewer 3 Report
Comments and Suggestions for Authors
Taking into account the supportive evidence provided by the authors (data on the CRES-T OsNAC60 transgenic line), the paper can be accepted for publication in its current form.